# Significant spatial patterns from the GCM seasonal forecasts of global precipitation

Tongtiegang Zhao[1], Wei Zhang[2], Yongyong Zhang[3], Zhiyong Liu[1], and Xiaohong Chen[1]

[1]Center of Water Resources and Environment, Southern Marine Science and Engineering Guangdong Laboratory (Zhuhai), School of Civil Engineering, Sun Yat-Sen University, Guangzhou 510275, China
[2]IIHR-Hydroscience & Engineering, University of Iowa, Iowa City 52242, United States
[3] Key Laboratory of Water Cycle and Related Land Surface Processes, Institute of Geographic Sciences and Natural Resources Research, Chinese Academy of Sciences, Beijing, 100101, China

*Correspondence to*: Tongtiegang Zhao (zhaottg@mail.sysu.edu.cn)

**Abstract:** Fully-coupled global climate models (GCMs) generate a vast amount of high-dimensional forecast data of the global climate; therefore, interpreting and understanding the predictive performance is a critical issue in applying GCM forecasts. Spatial plotting is a powerful tool to identify where forecasts perform well and where forecasts are not satisfactory. Here we build upon the spatial plotting of anomaly correlation between forecast ensemble mean and observations to derive significant spatial patterns to illustrate the predictive performance. For the anomaly correlation derived from the ten sets of forecasts archived in the North America Multi-Model Ensemble (NMME) experiment, the global and local Moran's I are calculated to associate anomaly correlation at neighbouring grid cells to one another. The global Moran's I associates anomaly correlation at the global scale and indicates that anomaly correlation at one grid cell relates significantly and positively to anomaly correlation at surrounding grid cells. The local Moran's I links anomaly correlation at one grid cell with its spatial lag and reveals clusters of grid cells with high, neutral, and low anomaly correlation. Overall, the forecasts produced by GCMs of similar settings and at the same climate center exhibit similar clustering of anomaly correlation. In the meantime, the forecasts in NMME show complementary performances. About 80% of grid cells across the globe fall into the cluster of high anomaly correlation under at least one of the ten sets of forecasts. While anomaly correlation exhibits substantial spatial variability, the clustering approach serves as a filter of noise to identify spatial patterns and yields insights into the predictive performance of GCM seasonal forecasts of global precipitation.

## 1 Introduction

Global climate models (GCMs) have been steadily improved over the past decades and are being employed by major climate centers around the world to generate operational long-range forecasts [Doblas-Reyes et al., 2013; Saha et al., 2014; Bauer et al., 2015; Hudson et al., 2017; Kushnir et al., 2019], providing physically-based forecasts in comparison to conventional statistical forecasts [Mason and Goddard, 2001; Wu et al., 2009; Schepen et al., 2012]. In particular, the fully-coupled GCMs assimilate world-wide observational information to predict the global hydrological cycle [Merryfield et al., 2013; Saha et al.,

2014; Jia et al., 2015]. Equipped with physical and dynamical laws, GCMs can potentially make forecasts of longer lead time and higher skill than statistical models [Kirtman et al., 2014; Becker et al., 2014; Chen et al., 2017]. In terms of computation, global climate forecasting is as complex as the simulation of the human brain and of the evolution of the early Universe [Bauer et al., 2015]. Advances in super-computing facilitate the forecasting and make GCM forecasts readily available for
hydrological, environmental, and agricultural modelling [Sheffield et al., 2014; Vecchi et al., 2014; Bellprat et al., 2019; Pappenberger et al., 2019; Zhao et al., 2019a].

GCMs generate a vast amount of high-dimensional forecast data, including retrospective forecasts of past climate and real-time forecasts [Kirtman et al., 2014; Saha et al., 2014; Jia et al., 2015]. Due to the complexity of atmospheric processes and model physics, the predictive performance of GCM forecasts is not uniform but varies considerably across the globe [Yuan et
al., 2013; Tian et al., 2017; Zhao et al., 2018]. Therefore, interpreting and understanding the predictive performance is a critical issue in the applications of GCM forecasts [Doblas-Reyes et al., 2013; Saha et al., 2014; Jia et al., 2015; Hudson et al., 2017; Wang et al., 2019a]. There are various metrics to verify the attributes of forecasts [Murphy, 1993]. For example, bias in percentage indicates the extent to which the forecasts are persistently higher, or lower, than the corresponding observations; probability integral transform (PIT) evaluates the reliability of the spread of ensemble forecasts in capturing the distribution
of observations; and the continuous ranked probability score (CRPS) is a probability-weighted measure of the errors of ensemble members in relation to the observations [Murphy, 1993; Hersbach, 2000; Gneiting et al., 2007; Tian et al., 2018; Wang et al., 2019b]. The anomaly correlation that indicates how well large (small) values of forecasts correspond to large (small) values of observations is one of the most popular metrics [e.g., Yuan et al., 2011; Saha et al., 2014; Crochemore et al., 2016; Hudson et al., 2017; Zhao et al., 2017a]. Compared to PIT that requires a diagnostic plot and CRPS that relies on
numerical integration, anomaly correlation is conceptually simple, easy to implement, and also robust to missing and censored values [Yuan et al., 2011; Luo et al., 2014; Slater et al., 2017].

Spatial plotting with latitude and longitude has been extensively used to handle the dimensionality for the verification of GCM forecasts [Kirtman et al., 2014; Hudson et al., 2017; Slater et al., 2017]. The fact that forecasts are commonly generated by GCMs as grid-based data makes spatial plotting a particular tool of choice for verification [Merryfield et al., 2013; Saha et al.,
2014; Jia et al., 2015]. As to anomaly correlation, spatial plotting overcomes tedious eyeball search by grid cell and is effective in locating where there is a good correspondence between forecasts and observations and where the correspondence is not satisfactory [Luo et al., 2013; Saha et al., 2014; Crochemore et al., 2016; Zhao et al., 2018, 2019b]. Similarly, spatial plotting applies to other verification metrics, such as bias and CRPS, and facilitates the examination of forecast attributes [Hersbach, 2000; Gneiting et al., 2007; Kirtman et al., 2014].

The extensive use of spatial plotting underlines the importance of testing the significance of spatial patterns. In spatial statistics, one of the fundamental issues is "are the spatial patterns displayed by the spatial plots significant in some sense and therefore worth interpreting?" [Cliff and Ord, 1981; Anselin, 1995; Getis, 2007]. However, the test of significance is commonly missing in the spatial plotting of GCM forecasts. In other words, verification metrics, such as anomaly correlation, are calculated for each grid cell and then shown as they are. To some extent, the interpretation of predictive performance depends on the color

schemes, which are selected subjectively to represent the scale of verification metrics. There is the first law of geography – "everything is related to everything else, but near things are more related than distant things" [Tobler, 1970]. As to spatial plotting, the indication is that when verifying forecasts at one grid cell, attention also needs to be paid to forecasts at surrounding grid cells. For anomaly correlation, a grid cell with high correlation between forecasts and observations can be surrounded by grid cells with similarly high correlation, or by grid cells with low correlation. In the former case, the grid cell

is located in a region where the GCM forecasts tend to perform well. But in the latter case, the high correlation can be a suspicious outlier. Moreover, previous studies observed grid cells with negative anomaly correlation, i.e., large (small) values of forecasts correspond to small (high) values of observations [Zhao et al., 2017b, 2018, 2019b]. In such a case, forecasts are cautiously wrong. Therefore, it is critical to characterize the different cases in spatial plotting and test whether the spatial patterns are significant and worth further attention.

In this paper, we are motivated to introduce spatial statistics [e.g., Di Luzio et al., 2008; Lu and Wong, 2008; Woldemeskel et al., 2013] to investigate the spatial plotting of anomaly correlation at the global scale. As will be shown later in this paper, the technique of spatial clustering facilitates the identification of significant patterns of high, neutral, and low anomaly correlation and provides an objective approach to interpreting the predictive performance of GCM forecasts. For the purpose of inter-comparison, the examination of significant patterns in spatial plotting has been conducted for ten sets of GCM seasonal

precipitation forecasts in the North American Multi-Model Ensemble (NMME) experiment [Kirtman et al., 2014; Ma et al., 2016; Zhang et al., 2017]. In the remainder of the paper, the dataset of GCM seasonal forecasts is illustrated in Section 2; the spatial clustering using global and local Moran's I is detailed in Section 3; the results of anomaly correlation at the global scale and its clustering are shown in Section 4; the discussion and conclusions are respectively presented in Sections 5 and 6.

## 2 Data description

The NMME builds on existing GCMs in North America to provide quality-controlled forecast data to the community of climate research and applications. More than ten sets of GCM precipitation forecasts have been spatially regridded and temporally aggregated to form a consistent dataset [Kirtman et al., 2014]. Each set of forecasts overall has 5 dimensions. They are 1) start time $s$, when the forecasts are initialised; 2) lead time $l$, whose unit is month for the forecasts; 3) ensemble member $n$, which is meant to represent forecast uncertainty; 4) latitude $y$; and 5) longitude $x$. Taking the precipitation forecasts of the Climate

Forecast System version 2 [CFSv2, Saha et al., 2014] in NMME as an example, $s$ is the beginning of each month and its value represents the number of months since January 1960; $l$ is 0, 1, …, 9, i.e., the forecasts are for month 0 head (current month), month 1 ahead, …, and month 9 ahead; $n$ is numbered from 1 to 24, i.e., 24 ensemble members; $y$ is from -90 to 90 while $x$ is from 0 to 359, i.e., the spatial resolution is 1 degree by 1 degree (approximately 100 kilometres). In the meantime, NMME provides precipitation observations corresponding to the forecasts. Specifically, the Climate Prediction Center (CPC)'s merged

analysis of precipitation [CMAP; Xie and Arkin, 1997; Xie et al., 2007], which is monthly, has been regridded to 1-degree resolution to verify GCM forecasts [Kirtman et al., 2014; Chen et al., 2017; Zhao et al., 2018].

**Table 1: Basic information on the ten sets of GCM forecasts from the NMME experiment**

| Climate Centre | GCM | Number of ensemble members | Lead time (month) |
|---|---|---|---|
| Canadian Meteorological Center (CMC) | Canadian coupled model version 3 (CanCM3) | 10 | 0–11 |
| | Canadian coupled model version 4 (CanCM4) | 10 | 0–11 |
| Center for Ocean-Land-Atmosphere Studies, Rosenstiel School of Marine and Atmospheric Science (COLA-RSMAS) | Community climate system model version 3 (CCSM3) | 6 | 0–11 |
| | Community climate system model version 4 (CCSM4) | 10 | 0–11 |
| Geophysical Fluid Dynamics Laboratory (GFDL) | Climate model version 2.1 (CM2p1) | 10 | 0–11 |
| | Climate model version 2.1 (CM2p1-aer04) | 10 | 0–11 |
| | Climate model version 2.5 with forecast-oriented low ocean resolution (CM2p5-FLOR-A06) | 12 | 0–11 |
| | Climate model version 2.5 with forecast-oriented low ocean resolution (CM2p5-FLOR-B01) | 12 | 0–11 |
| National Center for Atmospheric Research (NCAR) | Community earth system model version 1 (CESM1) | 10 | 0–11 |
| National Centers for Environmental Prediction (NCEP) | Climate forecast system version 2 (CFSv2) | 24 | 0–9 |

Ten sets of precipitation forecasts, as well as CMAP observations, in the NMME are downloaded from the International Research Institute at the Columbia University (https://iridl.ldeo.columbia.edu/SOURCES/.Models/.NMME/). Their retrospective forecasts are complete in the period from 1982 to 2010 [Merryfield et al., 2013; Saha et al., 2014; Jia et al., 2015]. In the meantime, their real-time forecasts are updated periodically in a slightly different setting; for example, CFSv2 forecasts are generated since January 2011 using initial conditions of the last 30 days, with 4 runs from each day

(https://www.cpc.ncep.noaa.gov/products/CFSv2/CFSv2_body.html). Basic information on the forecasts are provided in Table 1. In the analysis, the attention is paid to the retrospective forecasts

$$F_{GCM} = \left[ f_{s,l,n,y,x} \right]_{GCM} \tag{1}$$

In Eq. (1), $f$ represents forecast values that are specified by the 5 dimensions; $F$, which is the set of forecasts, is marked by the GCM that generates the forecasts. It is noted that in NMME, $F_{GCM}$ are raw forecasts generated by GCMs and are not bias-corrected or downscaled.

The observed precipitation corresponding to the forecasts is denoted as

$$O = \left[ o_{t,y,x} \right] \quad (t = s + l) \tag{2}$$

As shown in Eq. (2), the observation in total has three dimensions: time $t$, whose value is the addition of lead time $l$ to start time $s$ in the alignment of observations with forecasts; latitude $y$; and longitude $x$. It is pointed out that while $F$ differs by GCM, $O$ is the same across the ten sets of forecasts.

The start time $s$ in Eqs. (1) and (2) comprises year $k$, i.e., 1982, 1983, …, 2010, and month $m$, i.e., January, February, …, and December. The predictive performance of GCM forecasts exhibits seasonality [Zhao et al., 2017a, 2017b, 2018]. Accordingly, in the analysis, forecasts are selected by fixing $m$ while varying $k$, e.g., pooling forecasts initialised in June 1982, June 1983, …, June 2010. The anomaly correlation is calculated by relating forecasts to the corresponding observations

$$ r = \frac{\sum_k \left( rf_k - \overline{rf} \right)\left( ro_k - \overline{ro} \right)}{\sqrt{\sum_k \left( rf_k - \overline{rf} \right)^2} \sqrt{\sum_k \left( ro_k - \overline{ro} \right)^2}} \tag{3} $$

The above formulation deals with $k$ and omits other dimensions, including $m$, $l$, $y$ and $x$, for the sake of simplicity. In Eq. (3), $rf_k$ ($ro_k$) is the rank of year $k$'s forecast ensemble mean (observation) in the 29 years' ensemble mean (observations); and $\overline{rf}$ ($\overline{ro}$) is the mean value of $rf_k$ ($ro_k$). In general, the anomaly correlation characterises how well large (small) values of ensemble mean correspond to large (small) values of observations. Good (poor) correspondence makes $r$ tend towards 1 (–1).

With Eq. (3), the set of anomaly correlation between $F_{GCM}$ and $O$ is evaluated

$$ R_{GCM} = \left[ r_{m,l,y,x} \right]_{GCM} \tag{4} $$

In which $r$ and $R$ are respectively the correlation coefficients and the set of correlation. $R$, which differs by GCM, has four dimensions: 1) month $m$, which substitutes start time $s$ in Eq. (1); 2) lead time $l$; 3) latitude $y$; and 4) longitude $x$. Comparing Eq. (4) to Eq. (1), the dimension $n$ of ensemble member is eliminated since the forecast ensemble mean is taken in the calculation of anomaly correlation (Eq. 3).

For selected GCM forecasts in month $m$ and at lead time $l$, the anomaly correlation between ensemble mean and observation forms a two-dimensional array by latitude and longitude. Here, spatial plotting applies to the presentation of anomaly correlation at the global scale. Following Eq. (4), the set of anomaly correlation is denoted as

$$ R_{GCM,m,l} = \left[ r_{y,x} \right] \tag{5} $$

In Eq. (5), $y$ and $x$ specify the location of grid cells. Denoting grid cell as $i$, the subscripts of latitude $y$ and longitude $x$ are merged into $i$ for the purpose of simplicity

$$ R_{GCM,m,l} = \left[ r_i \right] \tag{6} $$

In which $r_i$ represents the anomaly correlation at grid cell $i$, of which the latitude is $y_i$ and the longitude is $x_i$.

## 3 Methods

The spatial plotting employs certain pre-selected colour schemes to represent the value of anomaly correlation and show the grid cell-wise anomaly correlation as it is [e.g., Yuan et al., 2011; Kirtman et al., 2014; Ma et al., 2016]. Spatial patterns that represent clusters of grid cells with high anomaly correlation have been observed and highlighted in peer studies [e.g., Saha et al., 2014; Jia et al., 2015; Slater et al., 2017]. The spatial clustering associates anomaly correlation at neighbouring grid cells to one another and tests the significance of the patterns by random permutation [Anselin, 1995, 2006; Rey and Anselin, 2010]. Following the standard formulations of spatial statistics, the global Moran's I is calculated to examine the association among anomaly correlation at the global scale

$$I = \frac{\frac{1}{\sum_{i=1}^{N}\sum_{j=1,j\neq i}^{N} w_{i,j}} \sum_{i=1}^{N}\sum_{j=1,j\neq i}^{N} w_{i,j}\left(r_i - \bar{r}\right)\left(r_j - \bar{r}\right)}{\frac{1}{N}\sum_{i=1}^{N}\left(r_i - \bar{r}\right)^2} \tag{7}$$

In which $N$ is the number of grid cells indexed by $i$ and $j$ across the globe; $\bar{r}$ is the mean value of anomaly correlation; and $w_{i,j}$ is the spatial weighting coefficient that usually decays with the distance between $i$ and $j$ [Miller, 2004; Hao et al., 2016; Schmal et al., 2017]. At the right-hand side of Eq. (7), the denominator is the variance of $r_i$ across all the grid cells; and the numerator is the spatially-weighted and -averaged covariance between $r_i$ and $r_j$. Generally, the value of the global Moran's I ranges from -1 to 1. The similarity (dissimilarity) of $r_i$ to the surrounding $r_j$ makes $I$ tend toward 1 (-1), while the random distribution of anomaly correlation makes $I$ close to 0.

The spatial weight $w_{i,j}$ plays an important part in the calculation of I [Rey and Anselin, 2010]. Following the inverse distance weighting (IDW) interpolation in geosciences [Di Luzio et al., 2008; Lu and Wong, 2008; Woldemeskel et al., 2013], $w_{i,j}$ is formulated as follows

$$w_{i,j} = \frac{1}{d(i,j)^2} \tag{8}$$

In which $d(i,j)$ is the Euclidean distance between grid cells $i$ and $j$, i.e., $d(i,j) = \sqrt{(x_i - x_j)^2 + (y_i - y_j)^2}$. In addition, the cut-off threshold for $d(i,j)$ is set as 10 degrees (approximately 1,000 kilometres) to reduce the computational burden. That is, $w_{i,j}$ is set as 0 if $d(i,j)$ exceeds 10.

Adding to the global Moran's I, the local Moran's I is obtained to test whether $r_i$ at a certain grid cell $i$ significantly relates to surrounding $r_j$ at the local scale [Anselin, 2006; Hao et al., 2016; Yuan et al., 2018]

$$I_i = \frac{\left(r_i - \bar{r}\right) \dfrac{\displaystyle\sum_{j=1, j\neq i}^{N} w_{i,j}\left(r_j - \bar{r}\right)}{\displaystyle\sum_{j=1, j\neq i}^{N} w_{i,j}}}{\dfrac{1}{N}\displaystyle\sum_{i=1}^{N}\left(r_i - \bar{r}\right)^2} \tag{9}$$

As shown in the above formulation, $I_i$ is positive when $r_i$ and the surrounding $r_j$ are similarly high, or similarly low. On the other hand, $I_i$ is negative when a high (low) value of $r_i$ correspond to low (high) values of neighbouring $r_j$. Also, $I_i$ can be close to zero when $r_i$ or the surrounding $r_j$ is close to the mean value. The significance of $I_i$ is tested by random permutations [Rey and Anselin, 2010]. For each permutation, the values of $r_j$ are randomly rearranged, and then the local Moran's I is re-calculated. The permutations obtained a reference distribution for $I_i$ under the null hypothesis of randomly distributed anomaly correlation

[Anselin, 1995, 2006; Rey and Anselin, 2010]. Given a significance level $\alpha$, the quantiles $I_{\alpha/2}$ and $I_{1-\alpha/2}$ are retrieved from the reference distribution. Therefore, the two-tailed test of $I_i$ along with the anomaly correlation $r_i$ facilitates spatial clustering and derives five cases:

$$case_i = \begin{cases} HH & (I_i > I_{1-\alpha/2}) \cup (r_i > \bar{r}) \\ HL & (I_i < I_{\alpha/2}) \cup (r_i > \bar{r}) \\ NS & (I_{\alpha/2} \leq I_i \leq I_{1-\alpha/2}) \\ LH & (I_i < I_{\alpha/2}) \cup (r_i < \bar{r}) \\ LL & (I_i > I_{1-\alpha/2}) \cup (r_i < \bar{r}) \end{cases} \tag{10}$$

As illustrated in Eq. (9), the first case HH, which is short for high-high, indicates that a high value of $r_i$ is surrounded by high values of $r_j$; the second case is HL – high-low – a high value of $r_i$ surrounded by low values of $r_j$; the third case is NS – not

significant – the local association of $r_i$ with surrounding $r_j$ is not significant; the fourth case is LH – low-high – a low value of $r_i$ surrounded by high values of $r_j$; and the fifth case is LL – low-low – a low value of $r_i$ surrounded by low values of $r_j$. In this way, the significance of patterns, which generally represent clusters of grid cells with high (low) anomaly correlation, is examined for spatial plotting of anomaly correlation. $\alpha$ is set to be 0.05 in this paper.

## 4 Results

The spatial clustering is performed for the anomaly correlation across the ten sets of forecasts in NMME. In the analysis, the attention is mainly paid to June, July, and August (JJA), which are generally boreal summer and Austral winter. Specifically, the start time of the forecasts is June, and the forecasts at the lead times of 0, 1, and 2 months are aggregated to form the seasonal forecasts. In the meantime, forecasts initialized in September of total precipitation in September, October, and

November (SON), forecasts initialized in December of total precipitation in December, (the next) January, and (the next February) (DJF), and forecasts initialized in March of total precipitation in March, April, and May (MAM) are also investigated, with the results presented in the supplementary material.

## 4.1 Anomaly correlation in JJA

The anomaly correlation between ensemble mean and observation is evaluated for the ten sets of seasonal precipitation forecasts. In Figure 1, the spatial plots employ a diverging red-blue colour scheme to represent the value of anomaly correlation. Red pixels indicate positive correlation, while blue pixels negative correlation. For each set of forecasts, many instances of red pixels can be observed. That is, forecasts exhibit promising performance with ensemble mean positively correlated with observation in many instances. Meanwhile, there also exist instances of blue pixels. In those instances, forecasts are generally not right because large (small) values of ensemble mean coincide with small (large) values of observation. While an inter-comparison of the ten sets of GCM forecasts in terms of anomaly correlation is presented in Figure 1, the anomaly correlation exhibits considerable spatial variability that hinders the analysis across the different sets of forecasts. As a result, it is none too easy to identify regions where the forecasts persistently exhibit promising predictive performance.

The first row of Figure 1 is for the forecasts generated by two Canadian GCMs. Although CanCM3 and CanCM4 share the ocean components and have slightly different atmospheric components [Merryfield et al., 2013], their anomaly correlation shows differences. For example, in Asia and Africa, the clusters of red pixels do not seem to overlap but differ instead; and in Australia, the anomaly correlation is high in Southeast Australia and part of Western Australia for CanCM3 while it is high in East Australia for CanCM4. These results are in accordance with a previous finding that CanCM3 and CanCM4 tend to complement each other [Merryfield et al., 2013]. The second row of Figure 1 shows the performance of two sets of forecasts by COLA-RSMAS GCMs. Complementary performance is no longer seen. Instead, CCSM4 forecasts show higher anomaly correlation and largely outperform CCSM3 forecasts in North and South America, Africa, and Australia. The outperformance can be attributed to the developments in ocean, atmospheric, and land components and the new coupling infrastructure of CCSM4 [Gent et al., 2011].

The third and fourth rows of Figure 1 are for the forecasts produced by four GFDL GCMs. In the third row, CM2p1 and CM2p1-aer04 forecasts seem to exhibit similar anomaly correlation, which tends to be high in Northeast South America, Western Africa, and Southeast Australia. In the fourth row, CM2p5-FLOR-A06 and CM2p5-FLOR-B01 forecasts show similarly high anomaly correlation in Northeast and Southeast South America, Northeast Australia and part of West Australia. On the other hand, the anomaly correlation differs from the CM2p1/CM2p1-aer04 forecasts to the CM2p5-FLOR-A06/CM2p5-FLOR-B01 forecasts. Jia et al. [2015] illustrated that CM2p5-FLOR GCMs have higher-resolution atmospheric and land components but coarser-resolution ocean components than CM2p1 GCMs. It is likely the changes in the setting of GCMs that lead to the difference in predictive performance. The fifth row of Figure 1 is for NCAR-CESM1 and NCEP-CFSv2 forecasts. Compared to CESM1 forecasts, CFSv2 forecasts tend to exhibit similar anomaly correlation in South America and show higher anomaly correlation in Asia, Africa and Australia.

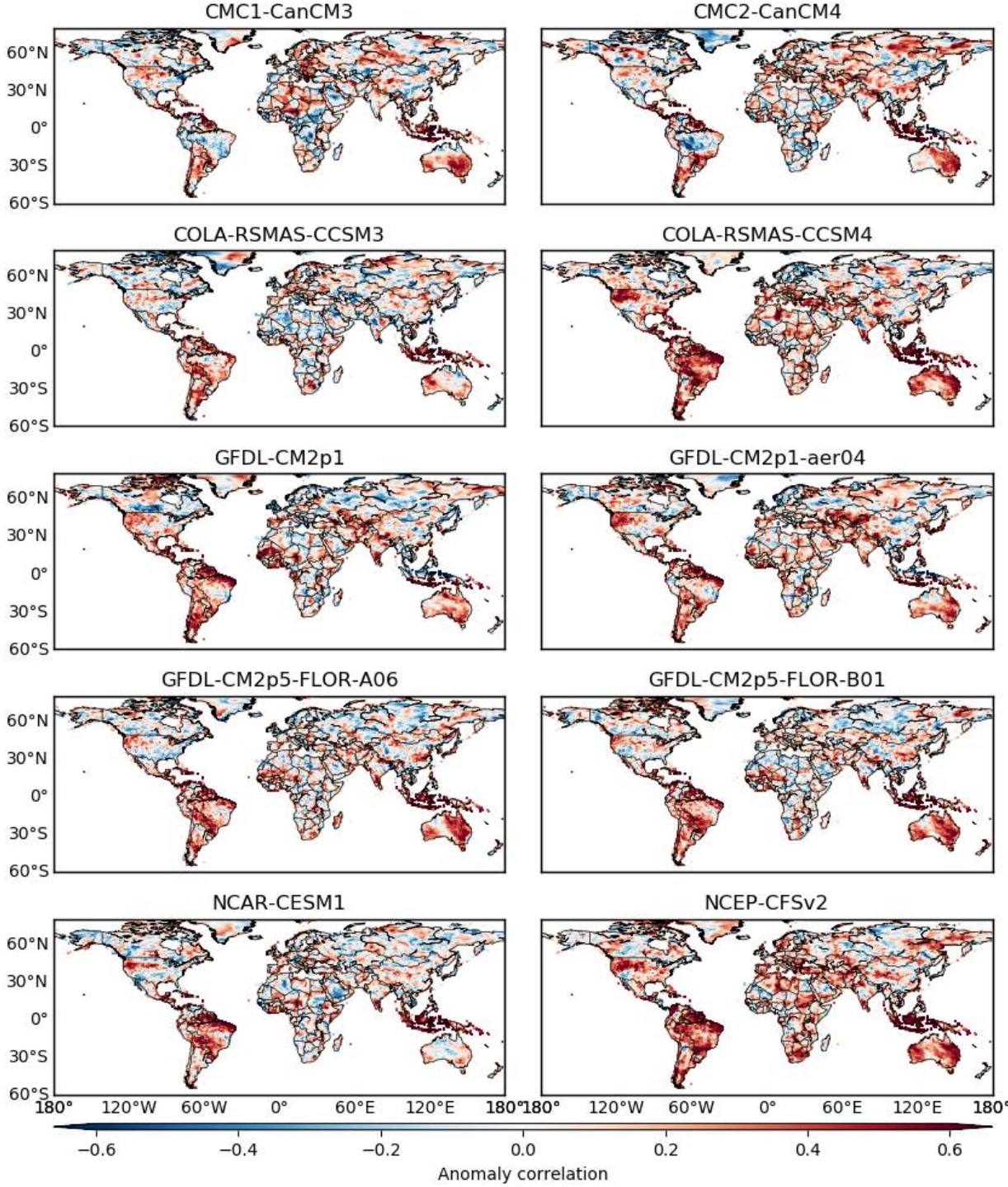

**Figure 1: Anomaly correlation between forecast ensemble mean and observation for ten sets of GCM forecasts of seasonal precipitation. The forecasts are initialized in June and are for the total precipitation in June, July, and August**

## 4.2 Anomaly correlation and its spatial lag in JJA

In spatial analysis, one critical issue is how an attribute at one location relates to the attribute at neighbouring locations [Cliff and Ord, 1981; Anselin, 1995; Getis, 2007]. For anomaly correlation, the subplots of Figure 1 imply the existence of some relationships as there are clusters of red pixels and of blue pixels. As for the clusters, Figure 2 presents a statistical test of the relationships using the global Moran's I. Specifically, for all the grid cells across the globe, the anomaly correlation at each grid cell is plotted against the spatially-weighted and -averaged anomaly correlation, i.e., spatial lag [Miller, 2004; Hao et al., 2016; Schmal et al., 2017], at the surrounding grid cells.

Figure 2 uses a viridis heatmap to indicate the density of scatter points. It can be observed that the points frequently fall in the first quadrant under all the ten sets of forecasts. In accordance with clusters of red pixels in Figure 1, this result suggests that many grid cells are with positive anomaly correlation and that they tend to be surrounded by grid cells with positive anomaly correlation. Meanwhile, some points are in the third quadrant. It is due to that some grid cells are of negative anomaly correlation and are surrounded by grid cells with negative anomaly correlation. This outcome corresponds to the existence of clusters of blue grid cells in Figure 1. Also, there are a few points in the second and fourth quadrants. Overall, anomaly correlation at one grid cell positively relates to anomaly correlation at the neighbouring grid cells. The global Moran's I is above 0.500, with the p-value far smaller than 0.01, for all the ten sets of NMME seasonal forecasts. Therefore, it is statistically verified that at the global scale, a grid cell with high (neutral, or low) anomaly correlation tends to be surrounded by grid cells with high (neutral, or low) anomaly correlation.

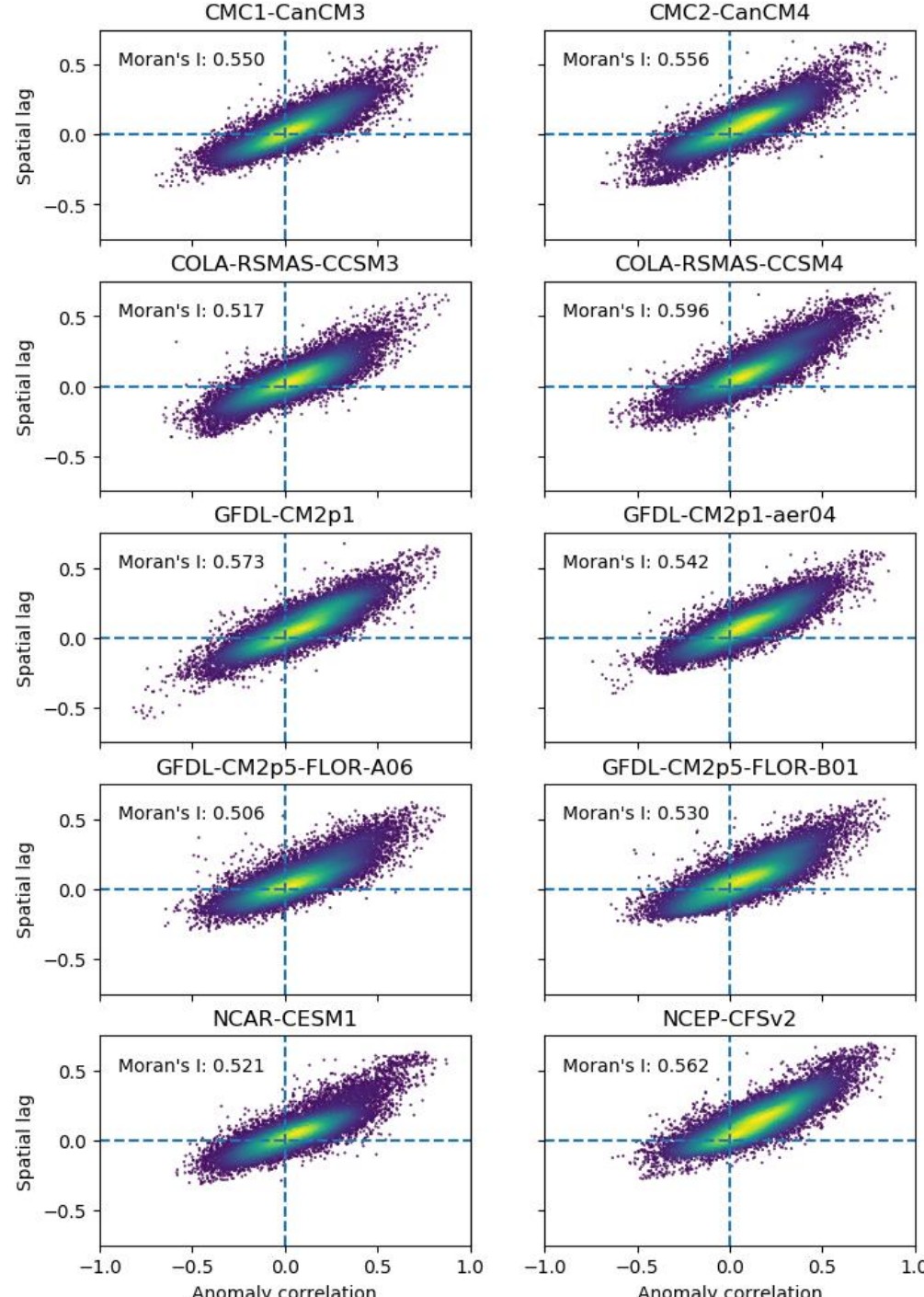

**Figure 2: Scatter plots of anomaly correlation at one grid cell against the corresponding spatial lag, i.e., spatially-weighted and -averaged anomaly correlation at surrounding grid cells. The density of points is estimated by kernel density function and shown by the viridis heatmap, with yellow (blue) colour indicating high (low) density**

## 4.3 Spatial clustering in JJA

Furthermore, the local Moran's I classifies the grid cells across the globe into 5 cases under each of the ten sets of forecasts.
In Figure 3, the five cases are marked by different colours. Specifically, the case HH is in orange, the case HL in red, the case NS in grey, the case LH in green, and the case LL in blue. A prominent finding from the subplots of Figure 3 is that the three cases of HH, NS, and LL have more instances than the other two cases of HL and LH. This result agrees to the spatial clustering of anomaly correlation in Figure 1 and to the distribution of scatter points in Figure 2. Comparing Figure 3 to Figure 1, it can be observed that orange regions generally correspond to clusters of red pixels, which represent positive anomaly correlation,
and that blue regions coincide with clusters of blue pixels, which show negative anomaly correlation. In the meantime, in-between orange and blue regions are grey regions. The implication is that regions with high and low anomaly correlation tend to be separated by regions with neutral anomaly correlation. While the spatial variability of anomaly correlation in Figure 1 is complicates the analysis of predictive performance, the classification in Figure 3 facilitates effective analysis across the ten sets of GCM forecasts.

The orange regions that correspond to clusters of grid cells with high anomaly correlation are of particular interest. Three findings are made from the spatial extent of orange regions. First of all, they tend to be similar under forecasts generated by the same climate center. For example, orange regions exist in a large part of South America for the ten sets of forecasts. On the other hand, they are not as extensive in the Amazon Basin under the CMC1-CanCM3 and CMC2-CanCM4 forecasts while they tend to cover Amazon under the COLA-RSMAS-CCSM3 and COLA-RSMAS-CCSM4 forecasts. The similarity versus
difference can be owing to that GCMs developed at the same climate center tend to share certain ocean, atmospheric, and land components [Gent et al., 2011; Merryfield et al., 2013; Jia et al., 2015]. Secondly, orange regions seem to be affected by the setting of GCMs. There are four sets of forecasts by GFDL. In Western United States, orange regions are extensive under the GFDL-CM2p1 and GFDL-CM2p1-aer04 forecasts but tend to be limited under the GFDL-CM2p5-FLOR-A06 and GFDL-CM2p5-FLOR-B01 forecasts. This drastic difference can be due to the setting of FLOR, i.e., forecast-oriented low ocean
resolution [Vecchi et al. 2014; Jia et al., 2015]. Thirdly, there are substantial regional variations possibly due to the predictability of seasonal precipitation [Doblas-Reyes et al., 2013; Becker et al., 2014; Zhang et al., 2017]. For example, orange regions cover large part of Australia, in particular Southwest and Southeast Australia. However, they are not as extensive in Europe, Asia and Africa. It is possibly owing to that the climate in Australia is strongly affected by ENSO [Schepen et al., 2012; Wang et al., 2012; Hudson et al., 2017] and that the 10 GCMs in NMME tend to capture the effect of ENSO on the total
precipitation in JJA.

The blue regions correspond to clusters of grid cells with low anomaly correlation. They are generally indicative of locations where forecasts are not satisfactory. Under the ten sets of forecasts, blue regions can be observed in large parts of Europe, Asia, Africa, Canada, and Eastern United States. While orange regions show some relationships with the source and setting of GCMs,

blue regions are more varying. In addition, they tend to mix with grey regions, which are indicative of neutral anomaly correlation, and also with red and green regions. Generally, this outcome implies the difficulty of generating skilful climate forecasts at the global scale as there are complex land-ocean-atmosphere processes [Bauer et al., 2015; Kapnick et al., 2018; Kushnir et al., 2019]. It is noted that some red regions that represent the case HL are observed to be located inside blue regions. The implication is that some grid cells may happen to exhibit high anomaly correlation but their surrounding grid cells are of low anomaly correlation. From the perspective of spatial statistics, the high correlation is not trustworthy and can be outlier.

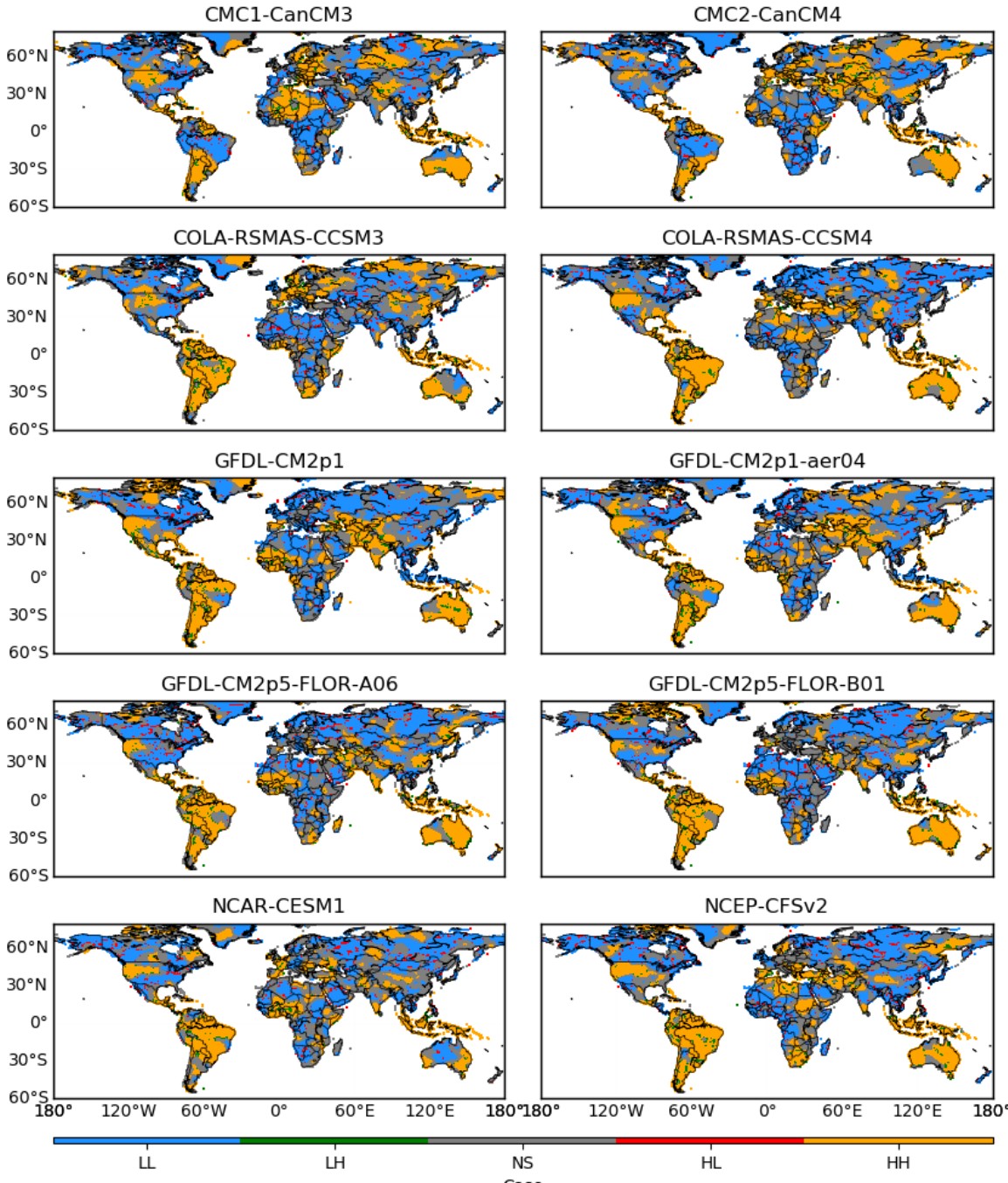

**Figure 3: Classification of grid cells across the globe into five cases based on spatial clustering of anomaly correlation. The case HH is marked in orange, the case HL in red, the case NS in grey, the case LH in green, and the case LL in blue. H and L are respectively short for high and low; the case HH (HL, LH, and LL) indicates that a grid cell with high (high, low, and low) anomaly correlation is surrounded by grid cells with high (low, high, and low) anomaly correlation. NS is short for not significant; the case NS means that the anomaly correlation at a grid cell or surrounding grid cells is neutral**

### 4.4 Frequency of the case HH in JJA

While the orange regions of the case HH are indicative of promising predictive performance, grid cells classified as this case differ across the ten sets of forecasts. To deal with the spatial variation of the case HH, the frequency that a grid cell falls into orange regions is counted for Figure 3. For one grid cell, the frequency ranges from 0 to 10. That is, across the 10 sets of forecasts, one grid cell is with high anomaly correlation and is surrounded by grid cells with high anomaly correlation at the minimum for 0 times and at the maximum for 10 times. Figures 4 and 5 illustrate the spatial and statistical distributions of the frequency, respectively.

Substantial regional variation can be observed for the frequency of the case HH from Figure 4. In North America, the frequency is evidently higher in Western United States than in Eastern United States, Canada, and Mexico. Globally, the frequency is higher in South America than in Europe, Asia, and Africa. Also, the frequency is high in Australia and Southeast Asia. Mason and Goddard [2001] elaborated on the relationship between ENSO and global seasonal precipitation anomalies: for the total precipitation in JJA, El Niño was shown to coincide with above-normal precipitation in parts of South and North America and below-normal precipitation in parts of Australia and Southeast Asia; by contrast, the impact of El Niño is not prominent for large parts of Europe, Asia, and Africa. With Mason and Goddard's finding, it is speculated that the results in Figure 4 to some extent reflect the impact of ENSO at the global scale.

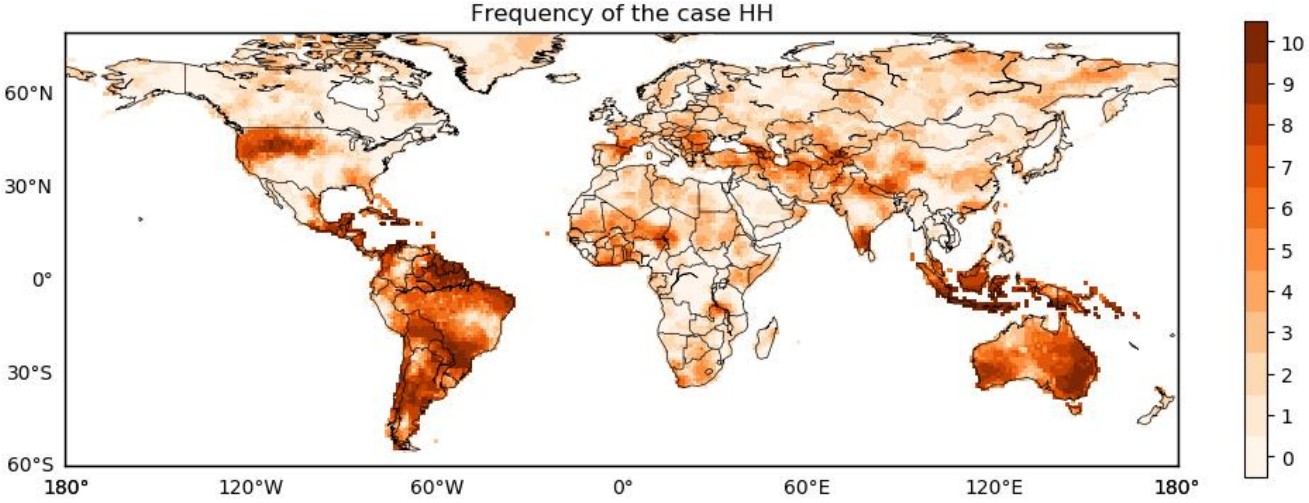

**Figure 4: The spatial distribution of the frequency of the case HH across the globe for the ten sets of GCM forecasts of the total precipitation in JJA**

The percentage and cumulative percentage of the frequency of the case HH are shown by bar and line plots in Figure 5, respectively. The frequency of 0 corresponds to a percentage of nearly 20%. This outcome means that about 20% of the grid cells across the globe do not fall into the case HH in any of the ten sets of forecasts. Another interpretation of this result is that about 80% of the grid cells fall into the case HH in at least one of the ten sets of forecasts. This result is in contrast to Figure 3 suggesting that orange regions are limited under each of the ten set of forecasts. It highlights the spatial complementarity among the multiple sets of GCM forecasts [Doblas-Reyes et al., 2013; Merryfield et al., 2013; Jia et al., 2015]. In the meantime, the percentages corresponding to the frequencies of 5, 6, …, 10 are all below 5% and the cumulative percentage reaches 80% at the frequency of 4. This result is due to that the performances of the different sets of forecasts are not the same. In other words, for certain regions, some sets of GCM forecasts may be not satisfactory while some other sets of GCM forecasts can be promising. Overall, Figure 5 suggests that GCM forecasts in NMME can complement each other [Wang et al., 2012; Becker et al., 2014; Kirtman et al., 2014].

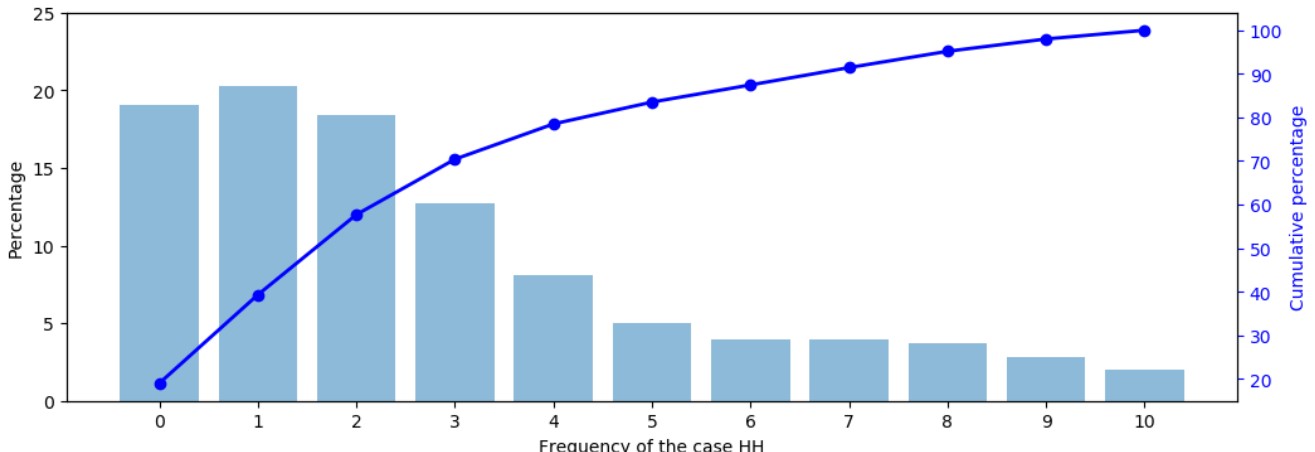

**Figure 5: Percentage (bar plot) and cumulative percentage (line plot) of the frequency of the case HH under the ten sets of GCM forecasts of the total precipitation in JJA**

### 4.5 Frequency of the case HH in SON, DJF, and MAM

Besides JJA, spatial clustering has been performed for the anomaly correlation of GCM seasonal forecasts of total precipitation in SON, DJF, and MAM. Similarly, it is observed that the anomaly correlation varies across the globe (Figures S1, S4, and S7 in the supplementary material), correlates with its spatial lag (Figures S2, S5, and S8), and exhibits significant spatial patterns (Figures S3, S6, and S9). In addition to Figures 4 and 5, the frequency of the case HH is counted for the other three seasons and shown in Figures 6 and 7.

ENSO is one of the most important drivers of global climate [Mason and Goddard, 2001; Saha et al., 2014; Bauer et al., 2015], and the CPC of NOAA has summarized the correlation between ENSO and global precipitation in different seasons (https://www.cpc.ncep.noaa.gov/products/precip/CWlink/ENSO/regressions/geplr.shtml). In this paper, the results in Figure 6 are associated with the global effects of ENSO. In SON, the CPC shows that ENSO correlates negatively with precipitation in Eastern Australia and Southeast Asia, and positively with precipitation in part of Middle East and East Africa. From the upper part of Figure 6, it is observed that the frequency of the case HH is high in these regions. In DJF, ENSO is shown to correlate positively with precipitation in Southern North America and negatively with precipitation in Northern South America. In these two regions, the frequency of the case HH is high (middle part of Figure 6). In MAM, ENSO is illustrated to correlate negatively with precipitation in part of Southeast Asia, Eastern Brazil, and Eastern Australia. Therein, the frequency of the case HH seem to be high (lower part of Figure 6). Therefore, as previous studies found that GCMs in NMME generate skilful forecasts of ENSO [e.g., Kirtman et al., 2014; Saha et al., 2014; Zhang et al., 2017], Figure 6 suggests that the skill, as is indicated by anomaly correlation, of GCM forecasts in NMME can also be related to ENSO. In Figure 7, the percentage and cumulative percentage of the frequency of the case HH are illustrated for SON, DJF, and MAM. Similar to Figure 5, the results show the complementarity among the ten sets of forecasts.

Besides ENSO, there are other drivers of global climate. For example, North Atlantic Oscillation (NAO) and Arctic Oscillation (AO) extensively affect the climate in Europe, Asia, and North America [Hurrell et al., 2001; Ambaum et al., 2002]. Several sea surface temperature indices of the Atlantic and Indian Oceans and ENSO jointly impact the climate in Africa [Rowell, 2013]. As can be observed from Figures 4, 5, 6, and 7, there is still substantial room for improvement of seasonal precipitation forecasts for large parts of Europe, Asia, and Africa. The overall neutrally skilful precipitation forecasts in these regions can possibly be due to that GCM formulations of other climate drivers are not as effective as the formulations of ENSO. In the meantime, the difficulty of global climate forecasting due to spatially-temporally varying teleconnections between regional precipitation and global climate drivers is noted [Merryfield et al., 2013; Saha et al., 2014; Jia et al., 2015; Hudson et al., 2017; Kushnir et al., 2019].

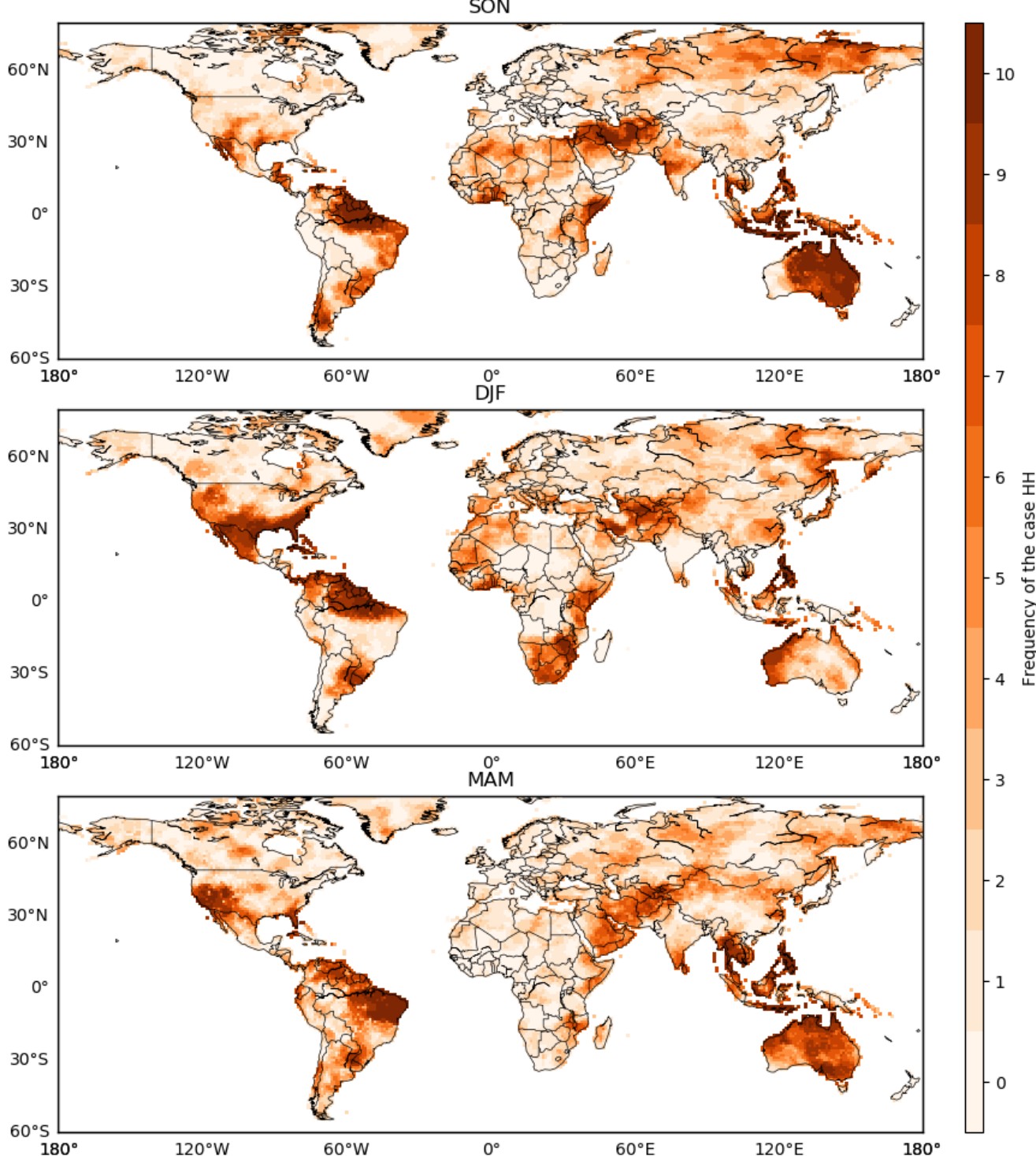

**Figure 6: As for Figure 4, but for SON, DJF, and MAM**

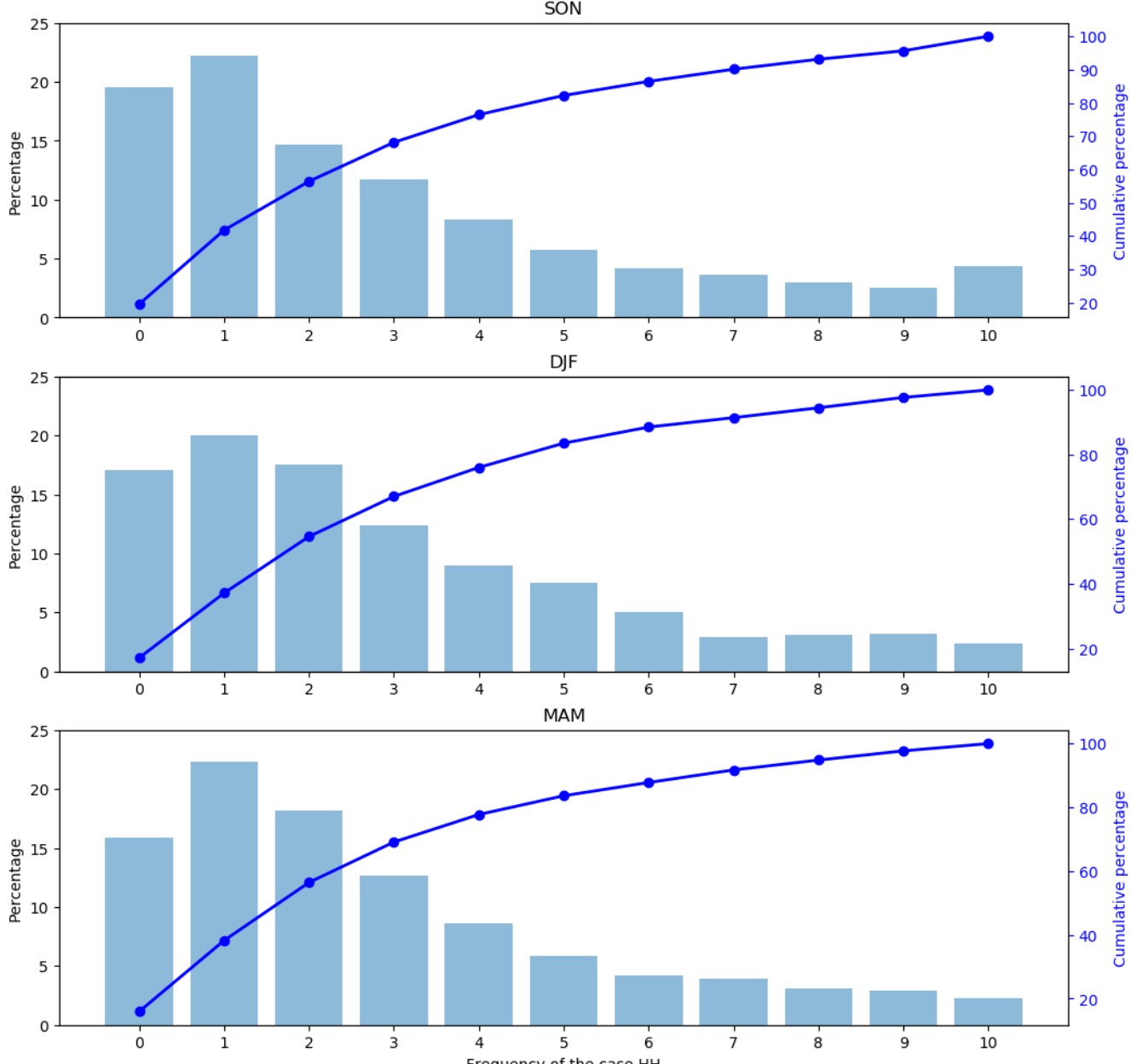

**Figure 7: As for Figure 5, but for SON, DJF, and MAM**

## 5 Discussion

This paper proposes to use spatial clustering to identify significant spatial patterns [Anselin, 1995; Miller, 2004; Schmal et al., 2017] from spatial plots of anomaly correlation, which have been widely used to illustrate the predictive performance of GCM forecasts. The test of significance is based on global and local Moran's I. The global Moran's I indicates that at the global scale anomaly correlation at one grid cell significantly relates to anomaly correlation at neighbouring grid cells, and the local Moran's I reveals clusters of grid cells with high anomaly correlation. Across the ten sets of GCM forecasts in NMME, the

clusters are observed in different regions across globe, which suggests that the skill of forecasts differs from region to region; in the meantime, the clusters vary by season owing to the seasonality of the skill of GCM forecasts [Doblas-Reyes et al., 2013; Becker et al., 2014; Yuan et al., 2015; Hudson et al., 2017; Kushnir et al., 2019]. To test whether the spatial patterns are robust, the observations of precipitation are also sourced from the Global Precipitation Climatology Centre (GPCC) [Becker et al., 2011; Schamm et al., 2014]. The anomaly correlation is re-calculated, and the spatial clustering is re-conducted. The results

of GPCC precipitation, which are shown in Figures S10 to 25 in the supplementary material, are overall similar to the results of CMAP precipitation. In particular, as to the two datasets of precipitation observations, the spatial distributions of the case HH resemble in JJA (Figures 4 and S10) and also in SON, DJF, and MAM (Figures 6 and S12). This outcome highlights the existence of significant spatial patterns and confirms that the spatial clustering can serve as an effective tool to yield insights into the predictive performance of GCM forecasts.

The spatial clustering ties anomaly correlation at neighbouring grid cells to one another and converts the continuous anomaly correlation into five categorical cases. Similar to the technique of moving average in time-series analysis, the categorical cases serve as a filter to reduce noise for the identification of spatial patterns. They handle the spatial variability of anomaly correlation and facilitate analysis across the ten sets of forecasts. It is illustrated that the forecasts produced by the same climate center tend to exhibit similar predictive performance and that changes in the setting of GCMs lead to changes in the predictive

performance. Given that the global and local Moran's I are flexible and easy to compute, they are ready to be extended in future analysis to other datasets of forecasts, such as forecasts generated by GCMs in Europe and Asia or by regional climate models (RCMs) [Alfieri et al., 2013; Bellprat et al., 2019; Kushnir et al., 2019]. Also, the forecasts can be verified using global and regional datasets of precipitation [Funk et al., 2015; Zhao et al., 2017a, 2017b]. A more extensive investigation would contribute to better understanding of the predictive performance and illustrate the advantages of different sets of forecasts. Of

particular interest is to explore which forecasts achieve promising predictive performance in large parts of Europe, Asia, and Africa. In the meantime, it is meaningful to account for the dynamics of global climate and investigate the model physics that leads to the improved performance.

The spatial clustering is a popular approach to geographical, ecological, and environmental modelling [e.g., Anselin, 1995, 2006; Miller, 2004; Hao et al., 2016; Schmal et al., 2017]. Meanwhile, its use appears to be not popular in the forecasting area.

A possible cause is that the objective of forecasting is usually location-specific. In other words, forecasts are produced for a certain site/watershed and then verified using the corresponding observations, of which the process does not involve other

sites/watersheds. In this paper, the analysis of GCM forecasts in NMME reveals that forecasts at neighbouring locations positively relate to one another. The indication is that the skill at one location can to some extent be inferred from adjacent locations. This result facilitates a new perspective for the verification of GCM forecasts. If a grid cell with high anomaly correlation is surrounded by grid cells with high anomaly correlation, the promising predictive performance at that grid cell can be confirmed. On the other hand, if the surrounding grid cells are with low, or even negative, anomaly correlation, then the high anomaly correlation is identified to be a suspicious outlier. Under that circumstance, further examination of the predictive performance is in demand to avoid undue optimism.

## 6 Conclusions

Fully-coupled GCMs perform physically-based forecasting of the global climate and generate a vast amount of spatial-temporal forecast data. The predictive performance is of both societal and scientific importance in the applications of these GCM forecasts. Focusing on the anomaly correlation between forecast ensemble mean and observation, we have conducted in-depth spatial analysis for ten sets of GCM forecasts in NMME and identified significant patterns from the spatial plotting of anomaly correlation. In the analysis of spatial clustering, grid cells across the globe are classified into five categories – HH, HL, NS, LH, and LL – depending on the anomaly correlation at that grid cell and the surrounding grid cells. The regions of grid cells with high, neutral, and low anomaly correlation are effectively identified. Further, effective inter-comparison across multiple sets of GCM forecasts is facilitated. While the analysis is concentrated on the spatial plotting of anomaly correlation, the framework readily applies to other metrics of GCM forecasts, such as bias, reliability, and skill. Moreover, the framework can be extended to GCM forecasts of other climate variables, for example temperature and wind speed, serving as a tool to explore GCM forecasts and interpret the predictive performance.

## Acknowledgements

The authors are grateful to the editor and the two anonymous reviewers for the constructive comments, which led to major improvements of the paper. The work is supported by the Natural Science Foundation of China (51979295, 51861125203, and U191120010), the Ministry of Science and Technology of China (2016YFC0400902 and 2017YFC0405900), and the Guangdong Provincial Department of Science and Technology (2019ZT08G090).

## Data availability

Both the forecasts and the observations can be downloaded from the International Research Institute for Climate and Society, Earth Institute, Columbia University (https://iridl.ldeo.columbia.edu/SOURCES/.Models/.NMME/)

## Competing interests

The authors declare that they have no conflict of interest.

## Author contributions

TZ, WZ, and YZ designed the experiment and performed the data analysis. TZ, ZL, and XC collected the data. TZ prepared the manuscript with contributions from all co-authors.

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
