# Peer review of "Significant spatial patterns from the GCM seasonal forecasts of global precipitation"

_Hydrology and Earth System Sciences, 2019_

## Referee Comment (RC1) · Anonymous Referee #1 · 7 Oct 2019

The authors used the spatial plotting method to identify significant patterns of fully-coupled GCMs from ten sets of NMME. Overall, the manuscript is well crafted with clear structures. Some grammatical errors exist and need careful proofreading. I have some minor comments.

1. Do you use some bias correction or downscale methods for the NMME forecasts? If yes, suggest to give some details.

2. Suggest to give the difference between the anomaly correlation and the spatial plotting method. Or the advantage of the spatial plotting method against the simple anomaly correlation.

3. Line 102: please check the start updated date of real-time forecasts.

[Figure]

4. Suggest to add the equation of the anomaly correlation calculation and give some details about the climatology for the anomaly.

5. Suggest to give some explanation for the forecasts of total precipitation in three months.

6. Are captions 4.1 and 4.2 the same? Please check.

7. Line 320: The authors showed the spatial extents of clusters vary by season. Suggest to give more explanations/reasons for this.

---

## Referee Comment (RC2) · Anonymous Referee #2 · 15 Oct 2019

This study examines the spatial patterns in skill of seasonal scale dynamical precipitation forecasts from the NMME models. By examining the spatial clusters of skill the study concludes that in general the skill pattern are spatially coherent – in other words regions with higher/positive [lower/negative] skills are typically surrounded by the pixels with higher/positive [lower/negative] skills. The study also finds the spatial pattern of the forecasts from the same climate center to be similar and that use of diverse models helps improve the regions with higher skill because of complementary skills.

This is a very useful study, as the authors argue that screening of forecast skill based on spatial pattern can help identify the regions with coherent high skill and hence the regions where the forecasts are likely to be most useful for decision-making. In addition to this implication, I think spatial pattern of regions with high skills can help with

the attribution of skill. For example, presumably, the regions with known ENSO tele-connection would show spatial coherence in high skill.

I think this study is certainly suitable for publication however I would like to suggest some additional analysis in the hopes of further improving this manuscript.

(1) The figure 4 and 5 are great as they summarize several relevant information on the skill for the JJA season, I think it would be good to add similar figures for other seasons here in the main manuscript rather than in supplementary material. As of now the main manuscript has only 5 figures so the manuscript certainly has space for it.

(2) It would also be instructive to examine how figure 4 and 5 change with increase in the lead-time, as in general, for decision-making applications forecasts are most useful at higher lead times. It would be very interesting to see how spatially coherent high skill regions changes at higher lead times and how complementary the forecasts from different models are in increasing the overall multimodel skill. It would also be interesting to look into the attribution of the high skill spatially coherent regions. Of course there are several sources of skill but at the least, I would suggest the authors to contrast Figure 4 (for at least DJF and JJA seasons) with ENSO and precipitation teleconnection maps (maps showing correlation between the two).

(3) The results of this study, I think can be used to also highlight the minimum number of models required for majority of the skill, which could be very useful for numerically expensive operational applications when impacts models (such as hydrologic or crop yield models) are driven by dynamical forecasts.

(4) Lastly, I can't help but wonder how the results of HH regions over Africa would vary if a different precipitation dataset such as the Climate Hazards Center Infrared Precipitation with Stations (CHIRPS, https://chc.ucsb.edu/data/chirps) dataset used. CHIRPS ingests a lot more local in-situ observations than other precipitation datasets largely based on global in-situ databases (e.g. GHCN), and hence tends to be better than other global datasets (see: https://www.nature.com/articles/sdata201566)
Minor comments:

(1) Abstract: Please briefly explain global and local Moran's in the abstract as well. (2) Figure 2 is missing a color bar. (3) Figure 3: Please provide legends explaining the abbreviations HH, HL etc.

---

## Author Comment (AC1) · 30 Oct 2019

*The authors used the spatial plotting method to identify significant patterns of fully coupled GCMs from ten sets of NMME. Overall, the manuscript is well crafted with clear structures.*

We appreciate the positive comments.

*Some grammatical errors exist and need careful proofreading. I have some minor comments.*

Thank you for the constructive comments. We have improved the paper accordingly and proofread the whole paper.

*1. Do you use some bias correction or downscale methods for the NMME forecasts? If yes, suggest to give some details.*

Thank you.

"In the analysis, the attention is paid to the retrospective forecasts

$$F_{GCM} = \left[ f_{s,l,n,y,x} \right]_{GCM} \qquad (1)$$

In Eq. (1), *f* represents forecast values that are specified by the 5 dimensions; *F*, which is the set of forecasts, is marked by the GCM that generates the forecasts. It is noted that in NMME, $F_{GCM}$ are raw forecasts generated by GCMs and are not bias-corrected or downscaled." (Page 4, Lines 103 to 107)

*2. Suggest to give the difference between the anomaly correlation and the spatial plotting method. Or the advantage of the spatial plotting method against the simple anomaly correlation.*

Thank you for the constructive comment. The advantage of spatial clustering is highlighted in the introduction:

"Spatial plotting with latitude and longitude has been extensively used to handle the dimensionality for the verification of GCM forecasts [Kirtman et al., 2014; Hudson et al., 2017; Slater et al., 2017]. The fact that forecasts are commonly generated by GCMs

as grid-based data makes spatial plotting a particular tool of choice for verification [Merryfield et al., 2013; Saha et al., 2014; Jia et al., 2015]. As to anomaly correlation, spatial plotting overcomes tedious eyeball search by grid cell and is effective in locating where there is a good correspondence between forecasts and observations and where the correspondence is not satisfactory [Luo et al., 2013; Saha et al., 2014; Crochemore et al., 2016; Zhao et al., 2018, 2019b]. Similarly, spatial plotting applies to other verification metrics, such as bias and CRPS, and facilitates the examination of forecast attributes [Hersbach, 2000; Gneiting et al., 2007; Kirtman et al., 2014].

The extensive use of spatial plotting underlines the importance of testing the significance of spatial patterns. In spatial statistics, one of the fundamental issues is "are the spatial patterns displayed by the spatial plots significant in some sense and therefore worth interpreting?" [Cliff and Ord, 1981; Anselin, 1995; Getis, 2007]. However, the test of significance is commonly missing in the spatial plotting of GCM forecasts. In other words, verification metrics, such as anomaly correlation, are calculated for each grid cell and then shown as they are. To some extent, the interpretation of predictive performance depends on the color schemes, which are selected subjectively to represent the scale of verification metrics. There is the first law of geography – "everything is related to everything else, but near things are more related than distant things" [Tobler, 1970]. As to spatial plotting, the indication is that when verifying forecasts at one grid cell, attention also needs to be paid to forecasts at surrounding grid cells. For anomaly correlation, a grid cell with high correlation between forecasts and observations can be surrounded by grid cells with similarly high correlation, or by grid cells with low correlation. In the former case, the grid cell is located in a region where the GCM forecasts tend to perform well. But in the latter case, the high correlation can be a suspicious outlier. Moreover, previous studies observed grid cells with negative anomaly correlation, i.e., large (small) values of forecasts correspond to small (high) values of observations [Zhao et al., 2017b, 2018, 2019b]. In such a case, forecasts are cautiously wrong. Therefore, it is critical to characterize the different cases in spatial plotting and test whether the spatial patterns are significant and worth further attention.

In this paper, we are motivated to introduce spatial statistics [e.g., Di Luzio et al., 2008; Lu and Wong, 2008; Woldemeskel et al., 2013] to investigate the spatial plotting of anomaly correlation at the global scale…" (Pages 2 to 3, Lines 50 to 74)

*3. Line 102: please check the start updated date of real-time forecasts.*

Thank you for the insightful comment. More details on real-time forecasts are added:

"Ten sets of precipitation forecasts, as well as CMAP observations, in the NMME are downloaded from the International Research Institute at the Columbia University (https://iridl.ldeo.columbia.edu/SOURCES/.Models/.NMME/). Their retrospective forecasts are complete in the period from 1982 to 2010 [Merryfield et al., 2013; Saha et al., 2014; Jia et al., 2015]. In the meantime, their real-time forecasts are updated periodically in a slightly different setting; for example, CFSv2 forecasts are generated since January 2011 using initial conditions of the last 30 days, with 4 runs from each day (https://www.cpc.ncep.noaa.gov/products/CFSv2/CFSv2_body.html))." (Page 4, Lines 98 to 103)

*4. Suggest to add the equation of the anomaly correlation calculation and give some details about the climatology for the anomaly.*

Thank you for the suggestion. We have added the equation:

"The start time *s* in Eqs. (1) and (2) comprises year *k*, i.e., 1982, 1983, …, 2010, and month *m*, i.e., January, February, …, and December. The predictive performance of GCM forecasts exhibits seasonality [Yuan et al., 2011; Zhao et al., 2017a, 2017b]. Accordingly, in the analysis, forecasts are selected by fixing *m* while varying *k*, e.g., pooling forecasts initialised in June 1982, June 1983, …, June 2010. The anomaly correlation is calculated by relating forecasts to the corresponding observations

$$r = \frac{\sum_k \left( rf_k - \overline{rf} \right)\left( ro_k - \overline{ro} \right)}{\sqrt{\sum_k \left( rf_k - \overline{rf} \right)^2} \sqrt{\sum_k \left( ro_k - \overline{ro} \right)^2}} \qquad (2)$$

The above formulation deals with *k* and omits other dimensions, including *m*, *l*, *y* and *x*, for the sake of simplicity. In Eq. (3), $rf_k$ ($ro_k$) is the rank of year *k*'s forecast ensemble mean (observation) in the 29 years' ensemble mean (observations); and $\overline{rf}$ ($\overline{ro}$) is the mean value of $rf_k$ ($ro_k$). In general, the anomaly correlation characterises how well large (small) values of ensemble mean correspond to large (small) values of observations. Good (poor) correspondence makes *r* tend towards 1 (–1)." (Page 5, Lines 112 to 119)

*5. Suggest to give some explanation for the forecasts of total precipitation in three months.*

Thank you. More information is provided:

"The spatial clustering is performed for the anomaly correlation across the ten sets of forecasts in NMME. In the analysis, the attention is mainly paid to June, July, and August (JJA), which are generally boreal summer and Austral winter. Specifically, the start time of the forecasts is June, and the forecasts at the lead times of 0, 1, and 2 months are aggregated to form the seasonal forecasts. In the meantime, forecasts initialized in September of total precipitation in September, October, and November (SON), forecasts initialized in December of total precipitation in December, (the next) January, and (the next February) (DJF), and forecasts initialized in March of total precipitation in March, April, and May (MAM) are also investigated, with the results presented in the supplementary material." (Pages 7 to 8, Lines 168 to 174)

*6. Are captions 4.1 and 4.2 the same? Please check.*

We are very sorry for the typo. The captions have been modified in the revision:

"4.1 Anomaly correlation in JJA" (Page 8, Line 175)

"4.2 Anomaly correlation and its spatial lag in JJA" (Page 10, Line 209)

*7. Line 320: The authors showed the spatial extents of clusters vary by season. Suggest to give more explanations/reasons for this.*

Thank you very much for the insightful comment. In the revision, we have added a new section and illustrated the results of the other three seasons:

"4.5 Frequency of the case HH in SON, DJF, and MAM

Besides JJA, spatial clustering has been performed for the anomaly correlation of GCM seasonal forecasts of total precipitation in SON, DJF, and MAM. Similarly, it is observed that the anomaly correlation varies across the globe (Figures S1, S4, and S7 in the supplementary material), correlates with its spatial lag (Figures S2, S5, and S8), and exhibits significant spatial patterns (Figures S3, S6, and S9). In addition to Figures 4 and 5, the frequency of the case HH is counted for the other three seasons and shown in Figures 6 and 7.

ENSO is one of the most important drivers of global climate [Mason and Goddard, 2001; Saha et al., 2014; Bauer et al., 2015], and the CPC of NOAA has summarized the correlation between ENSO and global precipitation in different seasons (https://www.cpc.ncep.noaa.gov/products/precip/CWlink/ENSO/regressions/geplr.sht ml). In this paper, the results in Figure 6 are associated with the global effects of ENSO.

In SON, the CPC shows that ENSO correlates negatively with precipitation in Eastern Australia and Southeast Asia, and positively with precipitation in part of Middle East and East Africa. From the upper part of Figure 6, it is observed that the frequency of the case HH is high in these regions. In DJF, ENSO is shown to correlate positively with precipitation in Southern North America and negatively with precipitation in Northern South America. In these two regions, the frequency of the case HH is high (middle part of Figure 6). In MAM, ENSO is illustrated to correlate negatively with precipitation in part of Southeast Asia, Eastern Brazil, and Eastern Australia. Therein, the frequency of the case HH seem to be high (lower part of Figure 6). Therefore, as previous studies found that GCMs in NMME generate skilful forecasts of ENSO [e.g., Kirtman et al., 2014; Saha et al., 2014; Zhang et al., 2017], Figure 6 suggests that the skill, as is indicated by anomaly correlation, of GCM forecasts in NMME can also be related to ENSO. In Figure 7, the percentage and cumulative percentage of the frequency of the case HH are illustrated for SON, DJF, and MAM. Similar to Figure 5, the results show the complementarity among the ten sets of forecasts.

Besides ENSO, there are other drivers of global climate. For example, North Atlantic Oscillation (NAO) and Arctic Oscillation (AO) extensively affect the climate in Europe, Asia, and North America [Hurrell et al., 2001; Ambaum et al., 2002]. Several sea surface temperature indices of the Atlantic and Indian Oceans and ENSO jointly impact the climate in Africa [Rowell, 2013]. As can be observed from Figures 4, 5, 6, and 7, there is still substantial room for improvement of seasonal precipitation forecasts for large parts of Europe, Asia, and Africa. The overall neutrally skilful precipitation forecasts in these regions can possibly be due to that GCM formulations of other climate drivers are not as effective as the formulations of ENSO. In the meantime, the difficulty of global climate forecasting due to spatially-temporally varying teleconnections between regional precipitation and global climate drivers is noted [Merryfield et al., 2013; Saha et al., 2014; Jia et al., 2015; Hudson et al., 2017; Kushnir et al., 2019].

[Figure]

Figure 6: As for Figure 4, but for SON, DJF, and MAM

[Figure]

Figure 7: As for Figure 5, but for SON, DJF, and MAM"

(Pages 16 to 19, Lines 311 to 345)

---

## Author Comment (AC2) · 30 Oct 2019

*This study examines the spatial patterns in skill of seasonal scale dynamical precipitation forecasts from the NMME models. By examining the spatial clusters of skill the study concludes that in general the skill pattern are spatially coherent – in other words regions with higher/positive [lower/negative] skills are typically surrounded by the pixels with higher/positive [lower/negative] skills. The study also finds the spatial pattern of the forecasts from the same climate center to be similar and that use of diverse models helps improve the regions with higher skill because of complementary skills.*

*This is a very useful study, as the authors argue that screening of forecast skill based on spatial pattern can help identify the regions with coherent high skill and hence the regions where the forecasts are likely to be most useful for decision-making. In addition to this implication, I think spatial pattern of regions with high skills can help with the attribution of skill. For example, presumably, the regions with known ENSO teleconnection would show spatial coherence in high skill.*

Thank you very much for the positive comments. We are encouraged by the comments and shall explore more in this area.

*I think this study is certainly suitable for publication however I would like to suggest some additional analysis in the hopes of further improving this manuscript.*

We are grateful to you for the constructive comments and have improved the paper accordingly.

*(1) The figure 4 and 5 are great as they summarize several relevant information on the skill for the JJA season, I think it would be good to add similar figures for other seasons here in the main manuscript rather than in supplementary material. As of now the main manuscript has only 5 figures so the manuscript certainly has space for it.*

Thank you very much for the insightful comment. We have added a new section and illustrated the results of the other three seasons:

"4.5 Frequency of the case HH in SON, DJF, and MAM

Besides JJA, spatial clustering has been performed for the anomaly correlation of GCM seasonal forecasts of total precipitation in SON, DJF, and MAM. Similarly, it is

observed that the anomaly correlation varies across the globe (Figures S1, S4, and S7 in the supplementary material), correlates with its spatial lag (Figures S2, S5, and S8), and exhibits significant spatial patterns (Figures S3, S6, and S9). In addition to Figures 4 and 5, the frequency of the case HH is counted for the other three seasons and shown in Figures 6 and 7.

ENSO is one of the most important drivers of global climate [Mason and Goddard, 2001; Saha et al., 2014; Bauer et al., 2015], and the CPC of NOAA has summarized the correlation between ENSO and global precipitation in different seasons (https://www.cpc.ncep.noaa.gov/products/precip/CWlink/ENSO/regressions/geplr.sht ml). In this paper, the results in Figure 6 are associated with the global effects of ENSO. In SON, the CPC shows that ENSO correlates negatively with precipitation in Eastern Australia and Southeast Asia, and positively with precipitation in part of Middle East and East Africa. From the upper part of Figure 6, it is observed that the frequency of the case HH is high in these regions. In DJF, ENSO is shown to correlate positively with precipitation in Southern North America and negatively with precipitation in Northern South America. In these two regions, the frequency of the case HH is high (middle part of Figure 6). In MAM, ENSO is illustrated to correlate negatively with precipitation in part of Southeast Asia, Eastern Brazil, and Eastern Australia. Therein, the frequency of the case HH seem to be high (lower part of Figure 6). Therefore, as previous studies found that GCMs in NMME generate skilful forecasts of ENSO [e.g., Kirtman et al., 2014; Saha et al., 2014; Zhang et al., 2017], Figure 6 suggests that the skill, as is indicated by anomaly correlation, of GCM forecasts in NMME can also be related to ENSO. In Figure 7, the percentage and cumulative percentage of the frequency of the case HH are illustrated for SON, DJF, and MAM. Similar to Figure 5, the results show the complementarity among the ten sets of forecasts.

Besides ENSO, there are other drivers of global climate. For example, North Atlantic Oscillation (NAO) and Arctic Oscillation (AO) extensively affect the climate in Europe, Asia, and North America [Hurrell et al., 2001; Ambaum et al., 2002]. Several sea surface temperature indices of the Atlantic and Indian Oceans and ENSO jointly impact the climate in Africa [Rowell, 2013]. As can be observed from Figures 4, 5, 6, and 7, there is still substantial room for improvement of seasonal precipitation forecasts for large parts of Europe, Asia, and Africa. The overall neutrally skilful precipitation forecasts in these regions can possibly be due to that GCM formulations of other climate drivers are not as effective as the formulations of ENSO. In the meantime, the difficulty of global climate forecasting due to spatially-temporally varying teleconnections between regional precipitation and global climate drivers is noted [Merryfield et al., 2013; Saha et al., 2014; Jia et al., 2015; Hudson et al., 2017; Kushnir et al., 2019].

[Figure]

Figure 6: As for Figure 4, but for SON, DJF, and MAM

[Figure]

Figure 7: As for Figure 5, but for SON, DJF, and MAM"

(Pages 16 to 19, Lines 311 to 345)

*(2) It would also be instructive to examine how figure 4 and 5 change with increase in the lead-time, as in general, for decision-making applications forecasts are most useful at higher lead times. It would be very interesting to see how spatially coherent high skill regions changes at higher lead times and how complementary the forecasts from different models are in increasing the overall multimodel skill. It would also be interesting to look into the attribution of the high skill spatially coherent regions. Of course there are several sources of skill but at the least, I would suggest the authors to contrast Figure 4 (for at least DJF and JJA seasons) with ENSO and precipitation teleconnection maps (maps showing correlation between the two).*

Thank you. The analysis is for seasonal forecasts that add monthly forecasts at three

lead times. The CPC of NOAA has investigated the global effects of ENSO and posted the plots of correlation on its website. We have contrasted the results of seasonal forecasts to the plots on CPC's website:

"ENSO is one of the most important drivers of global climate [Mason and Goddard, 2001; Saha et al., 2014; Bauer et al., 2015], and the CPC of NOAA has summarised the correlation between ENSO and global precipitation in different seasons (https://www.cpc.ncep.noaa.gov/products/precip/CWlink/ENSO/regressions/geplr.sht ml). In this paper, the results in Figure 6 are associated with the global effects of ENSO. In SON, the CPC shows that ENSO correlates negatively with precipitation in Eastern Australia and Southeast Asia, and positively with precipitation in part of Middle East and East Africa. From the upper part of Figure 6, it is observed that the frequency of the case HH is high in these regions. In DJF, ENSO is shown to correlate positively with precipitation in Southern North America and negatively with precipitation in Northern South America. In these two regions, the frequency of the case HH is high (middle part of Figure 6). In MAM, ENSO is illustrated to correlate negatively with precipitation in part of Southeast Asia, Eastern Brazil, and Eastern Australia. Therein, the frequency of the case HH seem to be high (lower part of Figure 6). Therefore, as previous studies found that GCMs in NMME generate skilful forecasts of ENSO [e.g., Kirtman et al., 2014; Saha et al., 2014; Zhang et al., 2017], Figure 6 suggests that the skill, as is indicated by anomaly correlation, of GCM forecasts in NMME can also be related to ENSO. In Figure 7, the percentage and cumulative percentage of the frequency of the case HH are illustrated for SON, DJF, and MAM. Similar to Figure 5, the results show the complementarity among the ten sets of forecasts." (Page 17, Lines 317 to 330)

*(3) The results of this study, I think can be used to also highlight the minimum number of models required for majority of the skill, which could be very useful for numerically expensive operational applications when impacts models (such as hydrologic or crop yield models) are driven by dynamical forecasts.*

Thank you. The performances of the GCM forecasts vary from region to region. This result suggests that different sets of forecasts can have advantages/disadvantages in different regions:

"While an inter-comparison of the ten sets of GCM forecasts in terms of anomaly correlation is presented in Figure 1, the anomaly correlation exhibits considerable spatial variability that hinders the analysis across the different sets of forecasts. As a result, it is none too easy to identify regions where the forecasts persistently exhibit

promising predictive performance.

The first row of Figure 1 is for the forecasts generated by two Canadian GCMs. Although CanCM3 and CanCM4 share the ocean components and have slightly different atmospheric components [Merryfield et al., 2013], their anomaly correlation shows differences. For example, in Asia and Africa, the clusters of red pixels do not seem to overlap but differ instead; and in Australia, the anomaly correlation is high in Southeast and part of Western Australia for CanCM3 while it is high in East Australia for CanCM4. These results are in accordance with a previous finding that CanCM3 and CanCM4 tend to complement each other [Merryfield et al., 2013]. The second row of Figure 1 shows the performance of two sets of forecasts by COLA-RSMAS GCMs. Complementary performance is no longer seen. Instead, CCSM4 forecasts show higher anomaly correlation and largely outperform CCSM3 forecasts in North and South America, Africa, and Australia. The outperformance can be attributed to the developments in ocean, atmospheric, and land components and the new coupling infrastructure of CCSM4 [Gent et al., 2011].

The third and fourth rows of Figure 1 are for the forecasts produced by four GFDL GCMs. In the third row, CM2p1 and CM2p1-aer04 forecasts seem to exhibit similar anomaly correlation, which tends to be high in Northeast South America, Western Africa, and Southeast Australia. In the fourth row, CM2p5-FLOR-A06 and CM2p5-FLOR-B01 forecasts show similarly high anomaly correlation in Northeast and Southeast South America, Northeast Australia and part of West Australia. On the other hand, the anomaly correlation differs from the CM2p1/CM2p1-aer04 forecasts to the CM2p5-FLOR-A06/CM2p5-FLOR-B01 forecasts. Jia et al. [2015] illustrated that CM2p5-FLOR GCMs have higher-resolution atmospheric and land components but coarser-resolution ocean components than CM2p1 GCMs. It is likely the changes in the setting of GCMs that lead to the difference in predictive performance. The fifth row of Figure 1 is for NCAR-CESM1 and NCEP-CFSv2 forecasts. Compared to CESM1 forecasts, CFSv2 forecasts tend to exhibit similar anomaly correlation in South America and show higher anomaly correlation in Asia, Africa and Australia." (Page 8, Lines 181 to 204)

*(4) Lastly, I can't help but wonder how the results of HH regions over Africa would vary if a different precipitation dataset such as the Climate Hazards Center Infrared Precipitation with Stations (CHIRPS, https://chc.ucsb.edu/data/chirps) dataset used. CHIRPS ingests a lot more local in-situ observations than other precipitation datasets largely based on global in-situ databases (e.g. GHCN), and hence tends to be better*

*than other global datasets (see: https://www.nature.com/articles/sdata201566)*

Thank you very much for the insightful comment. We agree with you on the importance of testing the results with another precipitation dataset. The CHIRPS dataset is of different spatial coverage and spatial-temporal resolution. In the revision, we have applied the Global Precipitation Climatology Centre (GPCC)'s monthly precipitation dataset to the test:

"To test whether the spatial patterns are robust, the observations of precipitation are also sourced from the Global Precipitation Climatology Centre (GPCC) [Becker et al., 2011; Schamm et al., 2014]. The anomaly correlation is re-calculated, and the spatial clustering is re-conducted. The results of GPCC precipitation, which are shown in Figures S10 to 25 in the supplementary material, are overall similar to the results of CMAP precipitation. In particular, as to the two datasets of precipitation observations, the spatial distributions of the case HH resemble in JJA (Figures 4 and S10) and also in SON, DJF, and MAM (Figures 6 and S12). This outcome highlights the existence of significant spatial patterns and confirms that the spatial clustering can serve as an effective tool to yield insights into the predictive performance of GCM forecasts." (Page 20, Lines 355 to 363)

[Figure]

Figure 4: The spatial distribution of the frequency of the case HH across the globe for the ten sets of GCM forecasts of the total precipitation in JJA

[Figure]

Figure S10: As for Figure 4, but for GPCC precipitation in JJA

[Figure]

Figure 6: As for Figure 4, but for SON, DJF, and MAM

[Figure]

Figure S12: As for Figure 6, but for GPCC precipitation in SON, DJF, and MAM

In the meantime, the importance of tests with more datasets is noted in the discussion:

"Given that the global and local Moran's I are flexible and easy to compute, they are ready to be extended in future analysis to other datasets of forecasts, such as forecasts generated by GCMs in Europe and Asia or by regional climate models (RCMs) [Alfieri et al., 2013; Bellprat et al., 2019; Kushnir et al., 2019]. Also, the forecasts can be verified using global and regional datasets of precipitation [Funk et al., 2015; Zhao et al., 2017a, 2017b]. A more extensive investigation would contribute to better understanding of the predictive performance and illustrate the advantages of different sets of forecasts. Of particular interest is to explore which forecasts achieve promising

predictive performance in large parts of Europe, Asia, and Africa. In the meantime, it is meaningful to account for the dynamics of global climate and investigate the model physics that leads to the improved performance." (Page 20, Lines 368 to 375)

*Minor comments:*

*(1) Abstract: Please briefly explain global and local Moran's in the abstract as well.*

Thank you.

"The global Moran's I associates anomaly correlation at neighbouring grid cells to one another and indicates that at the global scale anomaly correlation at one grid cell relates significantly and positively to anomaly correlation at surrounding grid cells. The local Moran's I links anomaly correlation at one grid cell with its spatial lag and reveals clusters of grid cells with high, neutral, and low anomaly correlation." (Page 1, Lines 14 to 17)

*(2) Figure 2 is missing a color bar.*

Thank you.

"Figure 2: Scatter plots of anomaly correlation at one grid cell against the corresponding spatial lag, i.e., spatially-weighted and -averaged anomaly correlation at surrounding grid cells. The density of points is estimated by kernel density function and shown by the viridis heatmap, with yellow (blue) colour indicating high (low) density" (Page 11, Lines 227 to 229)

*(3) Figure 3: Please provide legends explaining the abbreviations HH, HL etc.*

Thank you.

"Figure 3: Classification of grid cells across the globe into five cases based on spatial clustering of anomaly correlation. The case HH is marked in orange, the case HL in red, the case NS in grey, the case LH in green, and the case LL in blue. H and L are respectively short for high and low; the case HH (HL, LH, and LL) indicates that a grid cell with high (high, low, and low) anomaly correlation is surrounded by grid cells with high (low, high, and low) anomaly correlation. NS is short for not significant; the case NS means that the anomaly correlation at a grid cell or surrounding grid cells is neutral" (Page 14, Lines 269 to 273)